# Building consensus on priority areas for Sub-Saharan Africa's ageing population research: An e-Delphi study protocol

**Augustine Chukwuebuka Okoh**[1,2☯¶]*, **Ogochukwu Kelechi Onyeso**[1,3☯¶], **Wendy Ekemezie**[1‡¶], **Oluwagbemiga Oyinlola**[1,4,5‡¶], **Olayinka Akinrolie**[1,6☯¶], **Michael Kalu**[1,7☯¶], on behalf of the Emerging Researchers & Professionals in Ageing-African Network[¶]

1 Emerging Researchers & Professionals in Ageing–African Network, Abuja, Nigeria, 2 Health Research Methods, Evidence and Impact, McMaster University, Hamilton, Ontario, Canada, 3 Population Studies in Health, Faculty of Health Science, University of Lethbridge, Lethbridge, Alberta, Canada, 4 Medical Social Services Department, University College Hospital, Ibadan, Nigeria, 5 School of Social Work, McGill University, Montreal, Quebec, Canada, 6 Applied Health Science Program, Faculty of Graduate Studies, University of Manitoba, Winnipeg, Manitoba, Canada, 7 School of Kinesiology and Health Science, Faculty of Health, York University, Toronto, Ontario, Canada

☯ These authors contributed equally to this work.
‡ WE and OO also contributed equally to this work.
¶ Membership of the Emerging Researchers & Professionals in Ageing-African Network is provided in the Acknowledgments.
* okoha@mcmaster.ca

**Data Availability Statement:** No datasets were generated or analysed during the current study. All relevant data from this study will be made available upon study completion.

## Abstract

### Background

Improvement in medico-social services has increased life expectancy and population ageing in Sub-Saharan Africa (SSA). It was estimated that about 163 million people aged 65 and older will be resident in SSA by 2050. There is inadequate ageing research capacity in SSA which necessitates this study to (a) identify a decade-long ageing research opportunities, challenges, and solutions, and (b) prioritize critical ageing research areas and methodologies relevant to the SSA.

### Methods

We designed an e-Delphi protocol following the Reporting Guideline for Priority Setting of Health Research with Stakeholder. The stakeholders will be researchers, practitioners, older adults, and caregivers purposively selected through snowballing quota sampling to complete three rounds of e-Delphi surveys. Round 1 will involve open-ended questions derived from the study objectives. Responses from round 1 will be prepared as a checklist for stakeholders to rate during rounds 2 & 3, using a 9-point scale: low priority (1–3), moderate priority (4–6), and high priority (7–9). The criterion for reaching a consensus will be $\geq$ 70% of stakeholders rating an item "high priority" and $\leq$ 15% as "low priority." Quantitative data will be analysed using descriptive statistics, Wilcoxon matched-pairs signed-rank test will be used to assess the stability of stakeholders' responses, and qualitative comments will be analysed using content analysis.

**Funding:** The authors received no specific funding for this work.

**Competing interests:** The authors have declared that no competing interests exist.

## Discussion and implications

Setting aging research/practice priorities will help maximize the benefits of research investment and provide valuable direction for allocating public and private research funds to areas of strategic importance.

## Introduction

As life expectancy continues to increase in Sub-Saharan Africa (SSA), the population of older adults also grows, and aging research in the region is gaining attention [1]. The World Health Organization (WHO) projected the population of older adults in SSA to reach 67 million by 2025 and 163 million by 2050 [2]. This surge in the aging population has become a policy spotlight issue requiring prioritization of research interests and methodology to cater for this population's immediate needs [3]. Setting these priorities in aging research is crucial because it creates room for collaborative work among stakeholders where a collective understanding is reached about the most critical areas in research for older adults [4].

The World Health Organization's programme, The Decade of Healthy Ageing 2020–2030, promulgates preventive and remedial healthcare and social initiatives to mitigate the high burden of diseases and disabilities among older adults and support successful ageing [5]. Aging policies are becoming a mainstream discourse in many SSA countries, especially in South Africa, Kenya, Cameroon, Nigeria, Ghana, Mozambique, Uganda, Rwanda, Zimbabwe, Ethiopia, Malawi, and Tanzania [6]. The comprehensiveness of these countries' national ageing policies and their coverage and relevance to diverse groups of older adults remain unclear. For instance, some of these policies did not include or clarify the compromises expected for disadvantaged groups such as low-income households, rural dwellers, uneducated, and people with disabilities. One plausible reason could be a lack of research inclusiveness leaving out vital areas of inquiry, methodologies, demography, and voices. Thus, our study aspires to identify salient topic areas and relevant research methodologies that may address this concern.

Moreso, the dynamic nature of the social environment suggests a need to continuously update the policies to reflect older adults' prevailing state of affairs. For instance, research on health disparities has evolved in the past century from the conceptualization of poverty as a threshold, in gradients, as a mechanism, as having multiple levels of influence, to focusing on complex interactions among factors and causal pathways [7]. Also, the COVID-19 pandemic disrupted the pre-pandemic trajectory, increased pressure on the health system, widened health disparities, caused economic downturns, and contracted national budgets [8–10]. The conundrum of limited resources (workforce and funding) is even direr in low-income countries located in the SSA [10]. Hence, efficient allocation of scarce resources warrants the prioritization of issues and programs most relevant to older adults at this time.

Research is pivotal in agenda setting, developing evidence-informed policy alternatives, policy implementation, and evaluation [11]. Owing to the growing number of older adults, projected to be faster in SSA relative to the global north by 2050, the scope of contemporary aging research in SSA should be relevant to their health and social needs. Previous studies have contemplated key issues such as HIV/AIDs-related role reversal, intergenerational dependency, aging in place, access to health care, formal and informal social security systems, retirement homes, poverty, impacts of communicable and non-communicable, etc. [12]. Similarly, Kalu et al. conducted a systematic mapping review involving 512 ageing studies in SSA which showed that most studies focused on non-communicable diseases and HIV/AIDs, Cancer, and

physical functioning [see [3] for full details]. An update study conducted by Kalu et al. [13] assessed the quality of 544 articles with 25.9% being rated as fair or poor-quality. The authors suggested the need to diversify, prioritize, and improve the quality of ageing research in SSA.

The need to prioritize arises to help guide researchers, research funders, and policymakers in improving social policies, clinical guidelines and older patients' care [14]. With the limited research funding in the SSA for researchers and practitioners, it has become pertinent to set research priorities to enable researchers and practitioners to focus more on what matters the most to older adults in the SSA region. The Delphi approach offers an opportunity for consensus building on the research priorities. RAND corporation developed the Delphi approach in the 1950s to forecast the effect of technology on warfare, but over time, its application has moved health care and other fields. The Delphi method consists of panel selection, development of content surveys, and iterative stages to gain consensus [15]. It is a systematic approach to combining stakeholders' opinions–individuals with a range of knowledge, experience, and expertise on a topic through iterative, multi-stage completion of survey rounds until consensus is achieved [16].

This research will build on the findings of our previous studies [3,13,17]. This study will invite panellists, including researchers, policymakers, practitioners, older adults, and family caregivers, to participate in an international e-Delphi that aims to (a) identify and prioritise aging research topic areas and methodologies relevant to the SSA, (b) identify opportunities and challenges researchers faced in conducting ageing research in SSA, and (c) explore possible solutions to tackle the challenges researchers faced in conducting ageing research in SSA.

## Materials and methods

### Study design

The current e-Delphi study will adhere to reporting guideline for priority setting of health research (REPRISE) [18]. To breach the challenges of geographical spread of the anticipated stakeholders, we adopt the e-Delphi method involving rounds of web-based questionnaires [19], email exchanges between the principal researcher and the stakeholders when there is a need for clarification. The e-Delphi method will ensure that stakeholders are anonymous to each other, and that no stakeholder dominates the process, as noted in other methods for reaching consensus, such as Nominal Group [16]. A protocol paper for an e-Delphi study is recommended to promote data transparency, mitigate the risk of selective consensus reporting, and address the potential pitfalls associated with the Delphi approach [20].

### Stakeholders' description and eligibility

Stakeholders include: (i) gerontology researchers, (ii) practitioners, (iii) older adults, and (iv) non-medical caregivers in SSA. Stakeholders are individuals or groups who participate in or are affected by health- and healthcare-related decisions, programs, or policies [21]. To maximize the study's internal and external validities, stakeholders with expertise in the subject matter will be selected. Stakeholder selection will be based on specific qualifications, years of experience, or publications. The stakeholders will include (i) Researchers that have authored at least two peer-reviewed articles on older adults in SSA as either the first or senior author, (ii) Practitioners with at least two years of geriatric/gerontological experience in clinical or community health services in SSA, (iii) Older adults aged 55 years and older with self-identified lived experience of challenges of ageing in SSA, and (iv) Non-medical caregivers with at least one year of experience providing non-specialized paid or unpaid care to older adults in SSA.

Stakeholders are eligible to participate if they (i) conform with one of the expertise criteria, (ii) are fluent in the English language, (iii) are compliant with an online survey, and (iv) are

willing to provide consent and participate in the anticipate three rounds of the e-Delphi process [22].

## Sampling/recruitment technique

First, [researchers] stakeholders will be recruited purposively from a list of the SSA contacts we identified in the preceding phases of this four-part study [3,13,17]. We will invite the authors from the 544 articles included in our previous reviews [3,13,17]. Subsequently, snowballing method will be used until the desired sample size is reached [23]. We will encourage the stakeholders who were purposively selected to send us the name and email contact of other potential stakeholders who meet our inclusion criteria [19]. Regarding the recruitment of other stakeholders [healthcare practitioners, older adults, and non-medical caregivers], we will contact older people's homes, faith-based organizations, and geriatric organizations in SSA (e.g., South African Geriatrics Society). We will also contact academic organizations (e.g., Ibadan Longitudinal studies in aging centre, Centre for Ageing Research at the University of Ghana, Health and Aging in Africa: A Longitudinal Study of an INDEPTH Community [HAALSI], etc.), and other groups like HelpAge-Africa.

## Context

The SSA comprises 46 countries from the West, Central, East, and Southern Africa regions, excluding mainly North African countries [24]. Unlike the Arab-Asiatic North Africa, SSA is mostly inhabited by Black people and the categorization is based on race science and assume a homogeneity of Black experience [25]. Research priorities and interests tend to differ between North Africa and SSA because of perceived differences in challenges and dynamics as well as distinct historical background, political dynamics, and economic structures in these regions [26,27]. The SSA spans through several climatic zones, inhabited by people of diverse socio political and economic development, life expectancy, and multilingual cultures, speaking over 1000 languages. The United Nation's Department of Economic and Social Affairs, Population Division [28] estimated the population of SSA to reach 2.5 billion by 2050. We will recruit stakeholders from the SSA regions via sampling by convenience technique.

## Sample size

Keeney et al. [16] observed that there were no standard guidelines for e-Delphi sample size determination, and researchers reach a decision based on study objectives, scope, and methodology. A systematic review of previous e-Delphi studies reported sample size of 3 to 418, the median was 17 stakeholders [29]. The primary aim of sampling is to optimize research validity. We plan an e-Delphi that can reach data saturation (consensus) and validity using the least practicable number of stakeholders.

We will aim for a sample size between 60 to 120 stakeholders, with equal distribution across stakeholders. Using maximum demographic variation approach, we aim to recruit at least 16 stakeholders equally distributed across stakeholder groups: older adults (n = 4), caregivers (n = 4), gerontological practitioners and/or geriatricians (n = 4), gerontological researchers (n = 4) across from two countries (one low-income and one middle-income country) in the four regions of SSA–west Africa, central Africa, east Africa and southern Africa [24]. The Delphi method is prone to non-response or incomplete rounds, therefore Kalu et al. [19] recommended addition of about 25% of the anticipated sample size in readiness for the attrition. In our case, 25% of 64 = 16. Therefore, we will recruit an extra set of stakeholders from each of the SSA region, alternating low- and middle-income countries by means of balloting. Hence, the baseline sample size will be 80.

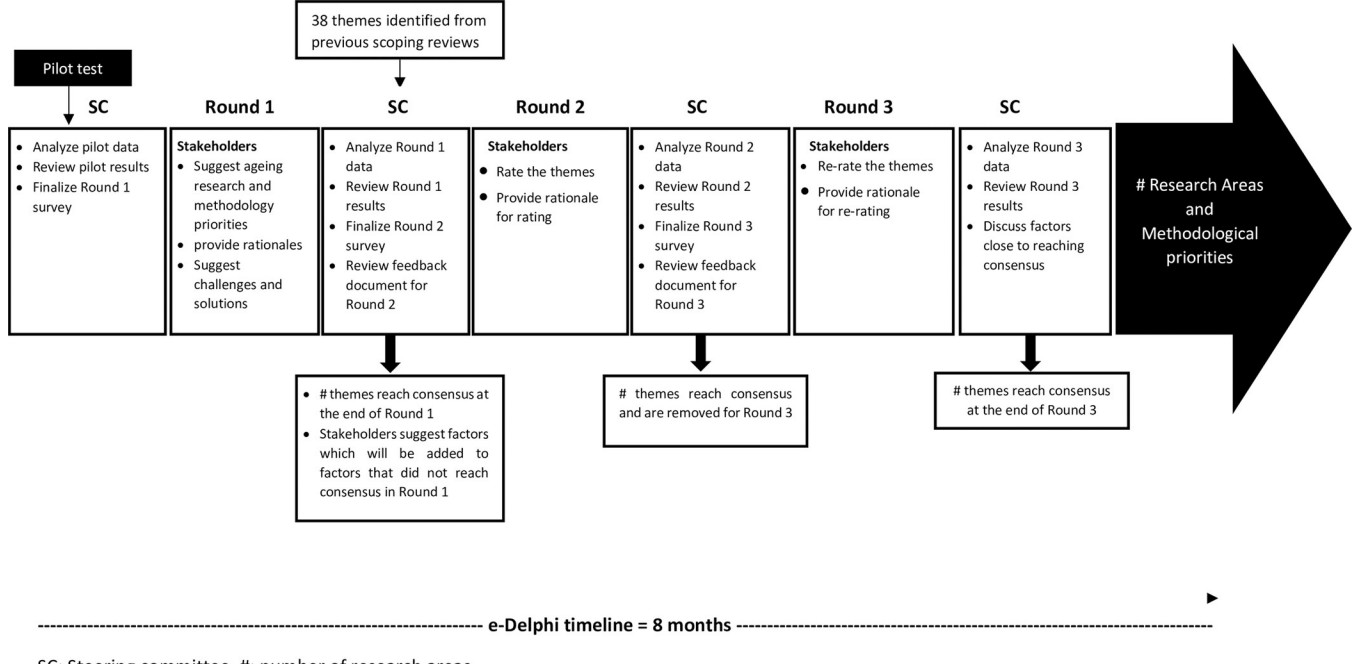

SC: Steering committee, #: number of research areas

**Fig 1. e-Delphi timeline, process, and results.**

## Steering committee

The Steering Committee (n = 10), including eight team investigators, an older adult, and a family caregiver, will provide overall study oversight. They include 8 investigators from the Emerging Researchers and Professionals in Aging-African Network (www.erpaan.org), an older adult, and a caregiver. The team investigators are physician (n = 1), social worker (n = 2), demographer (n = 1), physiotherapist-gerontologists (n = 2), nurse (n = 1), and optometrist (n = 1) researchers, with expertise in qualitative, quantitative, and Delphi methods. The older adults in the Steering Committee would have lived and worked in any of the SSA countries for at least five years. The caregiver would have at least five years of experience working in older peoples' homes or taking care of relatives who are older adults in family settings. The Steering Committee will meet at key stages throughout the study and will be responsible for identifying and responding to any issues arising, reviewing study conduct, and overseeing knowledge dissemination (see Fig 1). The Steering Committee will conduct content and face validity of e-Delphi questionnaire to ensure accuracy, comprehensiveness, clarity of wording, and appropriateness of structure; review results at each round; and review feedback summaries to be provided to stakeholders at subsequent rounds [19]. The Steering Committee can overrule items which the stakeholders could not reach a consensus; this approach reduces the burden on the stakeholders during the Delphi process [30].

## General procedure

**Number of rounds.** The number of rounds at which consensus is usually achieved in Delphi method is between two and four [23,31]. However, we anticipate that the current study will have up to three rounds due to the broadness of the topic and the sociodemographic heterogeneity of the potential stakeholders. It is noteworthy that multiple rounds increase the tendency of attrition and reduced response rate, which affects the Delphi process's validity [19].

**e-Delphi questionnaire.**    The questionnaire for round 1 of the e-Delphi survey will encompass the study objectives, description of e-Delphi methods, informed consent form, sociodemographic information of the stakeholders, open-ended questions regarding ageing research areas and methodologies, opportunities and challenges faced in conducting ageing research SSA, and potential solutions suggested by the stakeholders. Subsequent rounds of the survey will involve rating the priority level of responses from themes derived from open-ended questions in Round 1 and the identified ageing research areas from a systematized review [3] using a 9-point Likert scale: Not important (1–3), Important but not critical (4–6), and Critical (7–9). Stakeholders will also have the option to select "Unable to score" for any question they find uncomfortable to rate [19]. A 9-point scale is preferred in a Delphi survey as it increases sensitivity and consensus can be achieved on more items compared to 3- or 5-point scale [32].

**Survey administration platform.**    The survey will be administered through *DelphiManager©*, a web-based system designed to facilitate the building of e-Delphi surveys that includes functionality that allows for easy and efficient data management [33]. Delphi Manager allows for secure data collection and integrity using multiple encryption layers, and "quasi anonymity". Quasi-anonymity refers to researchers knowing the stakeholder's identity and their responses, but stakeholders and their responses are anonymous to each other [34]. Consenting stakeholders will be sent a weblink to an e-Delphi questionnaire. At the end of each round the stakeholders will be provided with a check box to indicate if they what to continue in the next round.

**Strategies to enhance result reliability and validity for use.**    We will invite older adults and caregivers as stakeholders, as the questions and definitions of terms and concepts will be provided in a plain language [19,35]. To ensure a comprehensive representation of opinions and minimize the risk of false consensus, stakeholders will be invited to participate in all rounds, irrespective of their previous round participation, as suggested in previous research [36]. While there is no set time limit in the literature, Delphi study duration ranges from 10 days to 10 weeks [31]. Based on previous international e-Delphi studies with multi-stakeholders [19,35], we estimate that each round will be open for four weeks to ensure stakeholders have enough time to participate effectively, without losing interest. There will be a minimum of four-week time interval between rounds to allow for data analysis, review of results by the Steering Committee, revision of the questionnaire and development of the feedback document for subsequent rounds [19]. The feedback document will include a summary document of preceding rounds, including bar graphs showing stakeholder group ratings, distribution of ratings, and individual ratings on factors that reached consensus and did not reach consensus and a qualitative summary of the rationale for each rating. This approach will improve transparency, validity and allows stakeholders to consider the group rating compared to their response enabling them to reflect, revise or change their opinions while avoiding any stakeholders dominating the process as seen in face-to-face, focus group discussions. Therefore, this e-Delphi process including the pilot round is estimated to be between nine and twelve months.

## Test round (pilot study)

To ensure rigour, examine the recruitment strategies, assess consent and face validity; we will pilot the first round of the e-Delphi process [37]. Stakeholders will be asked to provide feedback on the clarity, relevance, time taken and format of the survey questionnaire, and preferred method of rating. We will also examine the usability or barriers to use of e-Delphi process—several barriers to using internet and social media have been identified among older adults in SSA [38]. As recommended by McPherson et al. [37], we will use purposive sampling

to recruit four stakeholders (a researcher, a practitioner, an older adult, and a caregiver) for the pilot e-Delphi study.

## Strategies to increase the response rate

To improve response and retention rate, and reduced attrition rate; similar strategies utilized in previous e-Delphi studies [19,39] will be adopted for this present study to achieve a minimum response rate of 75%. Once we have identified the stakeholders, first, we will send a personalized email invitations to participate in the study and prior to each e-Delphi round. We will highlight the importance of the stakeholders to complete the survey to obtain a meaningful data. Second, we will send out a weekly reminder to the stakeholders to complete the survey, and finally, appreciation emails will be sent out once all the rounds have been completed.

## Consensus priority level

While there is no agreed threshold for defining consensus level in an e-Delphi study, it is usually recommended to state a priori criterion consensus level that it is clearly defined; and which is suitable for the aims and research questions of the e-Delphi study [31,40]. The most common definitions for consensus are percent agreement and proportion of rating within a range (restricted and unrestricted) [41]. Similar to a recent e-Delphi by Kalu et al. [19], we define priority consensus as follow: $\geq 70\%$ of stakeholders rated an ageing research area and methodology as "high priority" (scores $\geq 7$) and $\leq 15\%$ of the stakeholders rated a research area and methodology as "low priority" (scores $\leq 3$). An item that does not meet these criteria will be removed or refined as decided by the steering committee at the end of Round 3.

## Delphi Round 1

During Round 1 of the study, participants will be invited to respond to an open-ended question where they can describe their areas of interest and methodologies in aging research or practice that should be prioritized in preparation for the aging population in SSA. They will also be asked to identify and describe the challenges currently faced in aging research in SSA and propose potential solutions to address these challenges. We will conduct content analysis (described below) to create themes based on the stakeholders' responses in Round 1, which be added to the 38 research areas identified through our research team's systematized scoping review of 554 peer-reviewed aging articles published in SSA [3].

## Delphi Rounds 2 & 3

In Round 2 of the study, stakeholders will be requested to rate the priority level of the research themes and their corresponding methodologies from Round 1 using a 9-point Likert scale, ranging from "low priority (1–3)" to "moderate priority (4–6)" and "high priority (7–9)". Stakeholders can select "Unable to score" and provide rationales for their ratings of each research theme and methodology. Any items that reach a consensus priority at Round 2 will be removed to reduce the survey burden on the stakeholders and reduced lower response rate [42].

In Round 3, stakeholders will receive a summary of the Round 1 including items that reached consensus, those that did not reach consensus, and a summary of each stakeholder group rating for items, individual rating for each item. stakeholders will re-rate only items that has not reach consensus in Round 2 using the 9-point Likert scale, will reflecting on their initial response. Stakeholders will also be asked to provide rationale for re-rating the items. Stability testing will be conducted between items in Rounds 2 and 3. Only stable items that reached

consensus in Round 3 and those that had reached consensus in Round 2 would be included in the final research themes and methodologies priority list for aging research in SSA [43]. This iterative approach aims to facilitate the convergence of opinions among stakeholders and ensure a more refined consensus-building process by mitigating the potential coarseness of feedback received at each round [44].

At the end of each round, steering committee members will meet discuss the study analyzed and prepare the feedback documents for subsequent rounds.

## Data analysis

Quantitative data will be downloaded from the e-Delphi platform and processed using SPSS, version 28 (IBM Corp., Armonk, NY). Descriptive statistics including appropriate charts will be used to summarize and display the data. Obtained on the 9-point Likert scale, the data will be treated as ordinal continuous variables, where assumptions are met the parametric inferential statistics will be used for between and within round deviances in stakeholders' opinion. The response rate for each round will be calculated as the percentage of the number of usable responses returned relative to the total number of invitations. Median/mode, and percentages will be used as indicators of agreement on the 9-point scale, and consensus level, respectively [16]. The interquartile range and bar graphs will show the dispersion levels and individual ratings at each round for each factor. This will enable stakeholders to see where their responses stand in relation to the group's responses [19]. Stability, defined as the consistency of responses between successive rounds, will be used to assess the shift in scores across rounds because of considering the anonymized feedback from other stakeholders, and will be calculated using Cronbach alpha or intraclass correlation coefficient, if the data is normally distributed [45]. The analysis will be completed for the four categories of stakeholders, and regions of SSA.

NVivo Software© (version 12) will be used for qualitative (content analysis) analyses of stakeholders' comments in Round 1 and pertaining to the rationale of their choices at baseline and changes made in subsequent rounds [46]. Two coders will read the responses independently to develop codes and themes for each round. Coders will meet to merge their themes. Any disagreement will be discussed during a Steering Committee meeting and resolved. A reflexive journal and audit trail will be kept capturing methodological decisions throughout the Delphi process [19].

## Dissemination plan

The findings will be circulated among the Steering Committee Members for final feedback; afterwards the report will be sent to each of the stakeholders. The abstract and an infographic will be presented at two international ageing conferences focusing on SSA attendees and knowledge users. The findings will be documented following recent reporting guidelines for publication in an open-access international journal to promote visibility of the findings among researchers in SSA, enabling them to focus research on high-priority research topics and methods, which will improve the quality of ageing research into the future.

## Acknowledgments

We want to acknowledge other members of Emerging Researchers & Professionals in Ageing– African Network including Ebere Ugwuodo, Ezinne Ekediegwu, Ebuka Anieto, Blessing Ojembe, Anthony Iwuagwu, Michael Ibekaku, Deborah Adeleke, Henrietta Adandom, Ademuyiwa Adeboye, Chukwuebuka Okeke, Chiedozie Alumona, Chukwuebuka Onyekere, Miracle Ndubuaku, Chukwuenyegom Joseph Egbumike, Kelechi James Muomaife, Immaculata

Ugwuja, Kelechi Mirabel Onyeso, Emmanuela Aruma, John Osuolale Makanjuola, Funmibi Olatunji.

## Author Contributions

**Conceptualization:** Augustine Chukwuebuka Okoh, Michael Kalu.

**Methodology:** Augustine Chukwuebuka Okoh, Ogochukwu Kelechi Onyeso, Wendy Ekemezie, Oluwagbemiga Oyinlola, Olayinka Akinrolie, Michael Kalu.

**Project administration:** Augustine Chukwuebuka Okoh.

**Visualization:** Augustine Chukwuebuka Okoh.

**Writing – original draft:** Augustine Chukwuebuka Okoh, Ogochukwu Kelechi Onyeso, Wendy Ekemezie, Oluwagbemiga Oyinlola, Olayinka Akinrolie, Michael Kalu.

**Writing – review & editing:** Augustine Chukwuebuka Okoh, Ogochukwu Kelechi Onyeso, Olayinka Akinrolie, Michael Kalu.

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
