## [Editor Report · Decision Letter 0]

21 Nov 2023

PONE-D-23-20229Building consensus on priority areas for Sub-Saharan Africa’s ageing population research: An e-Delphi study protocolPLOS ONE

Dear Dr. Augustine Chukwuebuka Okoh,

Thank you for submitting your manuscript to PLOS ONE. After careful consideration, we feel that it has merit but does not fully meet PLOS ONE’s publication criteria as it currently stands. Therefore, we invite you to submit a revised version of the manuscript that addresses the points raised during the review process.

We look forward to receiving your revised manuscript.

Kind regards,

Naeem Mubarak, PhD

Academic Editor

PLOS ONE

Journal Requirements:

2. Please note that the Study Protocol article type is only suitable for proposals of studies that have not yet generated results. For further information, please see https://journals.plos.org/plosone/s/submission-guidelines#loc-study-protocols

Please update your Cover Letter to confirm that participant recruitment has not been completed.

Thank you for your attention. We look forward to hearing from you.

3. One of the noted authors is a group or consortium Emerging Researchers and Professionals in Ageing-African Network. In addition to naming the author group, please list the individual authors and affiliations within this group in the acknowledgments section of your manuscript. Please also indicate clearly a lead author for this group along with a contact email address.

4. Please amend your manuscript to include your abstract after the title page.

The protocol requires the following changes:

The manuscript aims to prioritize research areas for the aging population in SSA countries. The topic deserves appreciation being the least tapped research areas in low and middle-income countries itself. The protocol has merits to proceed further with a few suggestions.

1. In the background section, particularly in the second paragraph, there is a need to add more and updated references Also evidently state the research gap to comment on how your study is novel in context to low and middle-income countries.

2. The authors have stated the background of RAND cooperation and Delphi approach in the methodology section along with references while this should be mentioned only in the introduction part.

3. Please specifically explain the category of non-medical caregivers’ stakeholders concerning the areas of their expertise Also, mention the rationale for the exclusion of North African countries in your study.

4. Please include more clarity on the sampling technique used

5. Please use updated references for references # 5, 18, 29 and 38.

---

## [Author Response · Author response to Decision Letter 0]

9 Jan 2024

Changes in the protocol: 

1. In the background section, particularly in the second paragraph, there is a need to add more and updated references Also evidently state the research gap to comment on how your study is novel in context to low and middle-income countries.

We have provided the reference of a more WHO programme on healthy ageing. regarding the gap, this study intends to probe subject areas and methodologies that would advance inclusion of disadvantaged groups in research priorities for the LMICs. 

2. The authors have stated the background of RAND cooperation and Delphi approach in the methodology section along with references while this should be mentioned only in the introduction part.

The statements have been moved to the introduction section.

3. Please specifically explain the category of non-medical caregivers’ stakeholders concerning the areas of their expertise Also, mention the rationale for the exclusion of North African countries in your study.

unlike healthcare professionals, the non-medical caregivers imply individuals providing non-specialized care to an older adult. 

4. Please include more clarity on the sampling technique used

First, [researchers] stakeholders will be recruited purposively from a list of the SSA contacts we identified in the preceding phases of this four-part study (XX name blinded for peer review). We will invite the authors from the 544 articles included in our previous reviews (XX ref: blinded for peer review). Subsequently, snowballing method will be used until the desired sample size is reached [18]. We will encourage the stakeholders who were purposively selected to send us the name and email contact of other potential stakeholders who meet our inclusion criteria [14]. Regarding the recruitment of other stakeholders [healthcare practitioners, older adults, and non-medical caregivers], we will contact older people’s homes, faith-based organizations, and geriatric organizations in SSA (e.g., South African Geriatrics Society). We will also contact academic organizations (e.g., Ibadan Longitudinal studies in aging centre, Centre for Ageing Research at the University of Ghana, Health and Aging in Africa: A Longitudinal Study of an INDEPTH Community [HAALSI], etc.), and other groups like HelpAge-Africa.

5. Please use updated references for references # 5, 18, 29 and 38.

Previous (#5) Adler NE, Stewart J. Health disparities across the lifespan: Meaning, methods, and mechanisms. Ann N Y Acad Sci. 2010;1186: 5–23. doi:10.1111/J.1749-6632.2009.05337.X � I will keep this reference because it is still relevant and I did not find another relevant study that discussed the message I wanted to convey.

Previous (#18 - 19) Hasson F, Keeney S, McKenna H. Research guidelines for the Delphi survey technique. J Adv Nurs. 2000;32: 1008–1015. � Niederberger, M., & Spranger, J. (2020). Delphi technique in health sciences: a map. Frontiers in public health, 8, 457.

Previous (#29 - 33) Clibbens N, Walters S, Baird W. Delphi research: Issues raised by a pilot study. Nurse Res. 2012;19. doi:10.7748/nr2012.01.19.2.37.c8907 � McPherson, S., Reese, C., & Wendler, M. C. (2018). Methodology update: Delphi studies. Nursing research, 67(5), 404-410.

Previous (#38 - 42) Hsieh H-F, Shannon SE. Three Approaches to Qualitative Content Analysis. 2005. doi:10.1177/1049732305276687 � replaced with Kyngäs, H. (2020). Qualitative research and content analysis. The application of content analysis in nursing science research, 3-11.

*6. statement regarding the exclusion of North Africa added to the description of the research context

Unlike the Arab-Asiatic North Africa, SSA is mostly inhabited by Black people and the categorization is based on race science and assume a homogeneity of Black experience (Henn et al., 2012). Research priorities and interests tend to differ between North Africa and SSA because of perceived differences in challenges and dynamics as well as distinct historical background, political dynamics, and economic structures in these regions (Abebe & Ofosu-Kusi, 2016; Odigbo et al., 2023).

---

## [Editor Report · Decision Letter 1]

28 Jan 2024

Building consensus on priority areas for Sub-Saharan Africa’s ageing population research: An e-Delphi study protocol

PONE-D-23-20229R1

Dear Augustine Chukwuebuka Okoh,

We’re pleased to inform you that your manuscript has been judged scientifically suitable for publication and will be formally accepted for publication once it meets all outstanding technical requirements.

Kind regards,

Naeem Mubarak, PhD

Academic Editor

PLOS ONE

Additional Editor Comments (optional):

No further concerns to add. All the comments and suggestions have been completely addressed. Best of luck for your research ahead.
---

## [Editor Report · Acceptance letter]

12 Feb 2024

PONE-D-23-20229R1 

PLOS ONE

Dear Dr. Okoh, 

I'm pleased to inform you that your manuscript has been deemed suitable for publication in PLOS ONE. Congratulations! Your manuscript is now being handed over to our production team.

Kind regards, 

on behalf of

Dr Naeem Mubarak 

Academic Editor

PLOS ONE